# Genetic Testing Enhances the Precision Diagnosis and Treatment of Breast Cancer

**DOI:** 10.3390/ijms242316607

**Published:** 2023-11-22

**Authors:** Tinglin Yang, Wenhui Li, Tao Huang, Jun Zhou

**Affiliations:** Department of Breast and Thyroid Surgery, Union Hospital, Tongji Medical College, Huazhong University of Science and Technology, Wuhan 430022, Chinawenhui0415@163.com (W.L.)

**Keywords:** breast cancer, precision diagnosis, treatment, genetic testing, risk assessment

## Abstract

The contemporary comprehension of breast cancer has progressed to the molecular level. As a heterogeneous malignancy, conventional pathological diagnosis and histological classification could no longer meet the needs of precisely managing breast cancer. Genetic testing based on gene expression profiles and gene mutations has emerged and substantially contributed to the precise diagnosis and treatment of breast cancer. Multigene assays (MGAs) are explored for early-stage breast cancer patients, aiding the selection of adjuvant therapy and predicting prognosis. For metastatic breast cancer patients, testing specific genes indicates potentially effective antitumor agents. In this review, genetic testing in early-stage and metastatic breast cancer is summarized, as well as the advantages and challenges of genetic testing in breast cancer.

## 1. Introduction

Breast cancer ranks first in cancer incidence in the world. As the primary cause of cancer mortality in women, breast cancer has been one of the major threats to human health [1]. Currently, breast cancer is classified into different subtypes according to the expression levels of hormone receptors (HRs) estrogen receptor (ER) and progesterone receptor (PR), and human epidermal growth factor receptor-2 (HER-2) [2]. Different molecular subtypes of breast cancer exhibit unique biological behaviors and drug sensitivities. Although specific regimens are recommended for different subtypes of breast cancer in guidelines, the goal of precision medicine remains unrealized [3,4]. Intensive efforts have been put into managing breast cancer effectively and precisely, by which clinicians hope to improve the prognosis and enhance survival.

The comprehension of breast cancer at molecular levels deepened the cognition of key molecules and pathways that promote tumorigenesis. Genetic and genomic variation testing has now become an integral part of breast cancer management. Numerous clinical trials were set to assess different biomarkers in breast cancer so that patients could benefit from precise management. In addition to HRs and HER-2, progress has been achieved in the field of other biomarkers, such as *BRCA1/2*, *TP53*, and *PTEN* [5,6,7]. Based on single-gene biomarkers, the predictive value of different sets of genes was also explored with the higher pursuit of precise management and predicting prognosis. Moreover, contemporary techniques such as next-generation sequencing (NGS) have equipped clinicians and scientists with tools to carry out genetic testing and boosted the utilization of such testing in clinical practice [8].

Multigene assays (MGAs) and genetic testing for specific genes have been developed for patients with breast cancer (Figure 1). Selecting appropriate genetic testing based on the clinicopathological features of patients and comprehensively managing therapeutic regimens are gradually being promoted. This review illustrated representative oncogenic mutations and aberrant expressions of certain genes that reflect clinical outcomes, therapeutic effects, and malignant biological properties. For HR-positive breast cancer patients in the early stage, MGA including Oncotype Dx, MammaPrint, RecurIndex, breast cancer index (BCI), EndoPredict, and Prosigna Prediction Analysis of Microarray 50 (PAM50) are all helpful candidates for precisely predicting clinical outcomes [9,10,11,12,13,14,15]. For patients with metastatic breast cancer, genetic tests that include specific genetic mutations and biomarkers for immunotherapy also help to identify therapeutic targets and monitor efficacy [16,17,18]. In this review, different genetic tests for breast cancer are reviewed, as well as their advantages and limitations.

## 2. The Applications of Genetic Testing

The development of NGS and MGAs has boosted the promotion and application of genetic testing in clinical practice. Genetic testing is not only an important approach to molecular typing but also to finding therapeutic targets, predicting prognosis, and hereditary risks.

### 2.1. Evidence for Optimizing Regimens

With an in-depth understanding of tumorigenesis, tumor progression, and intrinsic signaling pathways of breast cancer, novel antitumor treatments have been explored. In breast cancer therapy, emerging molecular targeted agents have shown promising efficacy, including monoclonal antibodies or inhibitors targeting HER-2, phosphoinositide 3-kinase (PI3K) signaling pathway, cyclin-dependent kinases (CDKs), and poly (ADP-ribose) polymerase (PARP) [19,20,21,22,23,24,25]. In addition, immunotherapy, especially immune checkpoint inhibitors (ICIs), has demonstrated advancement in generating antitumor immune responses [26,27]. Genetic testing serves as a robust and efficient auxiliary examination that provides clinicians with molecular subtypes and therapeutic targets, indicating potential treatment strategies and drug selection. The detection of key molecules in the progression of breast cancer via genetic testing can be solid evidence for the enhancement of precision medicine.

Patients with advanced or metastatic breast cancers are frequently advised to undergo genetic testing. Despite curing advanced breast cancers remains challenging, the time and quality of survival can be improved by implementing individualized and optimized regimens [28]. Genetic testing has proven useful for indicating drug susceptibility and resistance in these patients, thereby screening proper and precise molecular targets. Genetic testing can offer benefits to breast cancer patients experiencing disease progression during treatment, whether undergoing testing for the first time or retesting [29,30]. Such testing can reveal drug-resistant mutations and newly exposed targets, enabling the evaluation of potential regimen substitutions.

### 2.2. Predictive Implication for Prognosis

A series of MGAs have been developed and commercially used for breast cancer patients. These assays combine genes that are closely related to tumor proliferation, tumor invasion, antitumor immunity, inflammation, and internal control genes. A score will be calculated by comprehensively analyzing the results of assays. According to the oncogenes in MGAs, predictive or prognostic information, including the risk of recurrence and the benefit of chemotherapy, can be generated. Genetic testing contributes to the selection of surgical procedures, neoadjuvant therapy, and adjuvant therapy for patients with unclear treatment options under the guidance of traditional clinicopathologic features [31,32]. In such cases, MGAs that are appropriate for patients need to be carefully chosen [33]. In addition, evidence from genetic tests and evidence from clinicopathologic features need to be analyzed and judged comprehensively.

### 2.3. Assessment of Hereditary Risks

Familial breast cancer was first observed and documented in the 1860s, and subsequent research has confirmed a link between family history and breast cancer [34,35]. Familial breast cancer is characterized by an earlier age of onset, autosomal dominant, and bilateral breasts involved [36]. Hereditary factors play important roles in the development of breast cancer, accounting for approximately 5–10% of cases. Germline mutations of *BRCA1/2* are the most concerned variants responsible for driving the susceptibility of familial breast cancer and ovarian cancer, collectively known as hereditary breast and ovarian cancers (HBOCs) [37]. Further studies unveiled other inherited protooncogenes that increase the risk of breast cancer [38,39,40]. Mutated *TP53* in Li-Fraumeni syndrome, mutated *PTEN* in Cowden’s syndrome and *PTEN* hamartoma tumor syndrome, and mutated *STK11* in Peutz–Jegher syndrome can increase the risk of breast cancer to around 50% [41,42,43]. In addition, mutations in *CDH1*, *ATM*, *CHEK2*, and *PALB2* are all oncogenic driver mutations. Notably, these mutations not only increase the risk of breast cancer but also the risk of multiple malignancies, including thyroid cancer, ovarian cancer, prostate cancer, sarcoma, and gastrointestinal cancer [44,45,46,47]. This detection offers importance to the early diagnosis, treatment, and detection of inherited breast cancers and other tumors. In addition, genetic testing provides a more explicit basis for the prevention of familial breast cancer at the genetic level.

## 3. Genetic Testing for Early-Stage Breast Cancer

Early-stage breast cancer refers to breast cancer at clinical stage I–stage II, accounting for 73.1% of breast cancer [48]. For patients at an early stage, rational and effective treatment is vital to improve prognosis. An increasing number of studies have made important progress, such as the OlympiA trial of adjuvant olaparib in patients with germline pathogenic variants in *BRCA1/2* [49] and the KARISMA trial of *CYP2D6* mutations in patients using tamoxifen [50]. Regimens of adjuvant and neoadjuvant treatments used to be mainly based on the clinicopathological characteristics of patients, but deficiencies are still noticed in forecasting the efficacy and prognosis. To complement the deficiency, MGAs, including Oncotype Dx, MammaPrint, RecurIndex, BCI, EndoPredict, and PAM50, were developed, as the following describes.

### 3.1. Oncotype Dx

Oncotype Dx, which is also called the 21-gene recurrence score (RS) assay, is an MGA for HR-positive HER-2-negative breast cancer patients that evaluates RS. Its capacity to predict prognosis and benefit from adjuvant chemotherapy or radiotherapy has been explored in breast cancer patients at stages N0–N1. The Oncotype Dx is relatively widely used in Europe and America. The 21 genes involved in Oncotype Dx were established from the NSABP B-14 study, which investigated the expression of 250 genes in 666 HR-positive node-negative breast cancer patients in the early stage. The risk of recurrence was initially divided into low (RS < 18), intermediate (18 ≤ RS < 31), and high (RS ≥ 31), according to RS [51]. Since RS proved to be an independent risk factor for the prognosis of HR-positive breast cancer patients, further study NSABP B-20 aimed to validate the predictive value of Oncotype Dx in chemotherapy sensitivity. Patients in the high-risk group tend to benefit from adjuvant chemotherapy, while patients in the low-risk group are deemed to have little benefit from chemotherapy. Although the benefit of chemotherapy is unclear for patients in the intermediate group, these patients may also benefit from chemotherapy in clinical practice [51,52]. The phase III trial TAILORx confirmed the value of predicting the efficacy of chemotherapy and prognosis that Oncotype Dx possesses. The risk threshold was reclassified as low risk (RS < 11), intermediate risk (11 ≤ RS < 26), and high risk (RS ≥ 26). HR-positive breast cancer patients in the intermediate-risk group were randomized to receive either endocrine therapy or endocrine therapy plus chemotherapy. There was no significant difference in invasive disease-free survival (iDFS) and overall survival (OS) between the endocrine therapy group and the chemoendocrine therapy group. However, adjuvant chemotherapy markedly reduced 9-year distant recurrence in patients younger than 50 years old. Adjuvant chemotherapy decreased rates of distant recurrence in patients with RS of 16 to 20 and 21 to 25 by 1.6% and 6.5%, respectively. Therefore, for patients in the intermediate-risk group, those who are younger than 50 and with RS ranging from 16 to 25 are more likely to benefit from chemoendocrine therapy [53,54].

The prognostic significance of Oncotype Dx for HR-positive breast cancer patients with lymph node metastasis was also explored by some clinical trials. The SWOG S8814 study analyzed 367 postmenopausal patients and revealed that higher RS was linked with a worse prognosis for patients who received endocrine therapy only. In addition, high-risk patients defined by RS could benefit from chemotherapy [55]. Another phase III study, RxPONDER, also investigated Oncotype Dx in HR-positive patients at stage N1 with RS less than 25. Patients received endocrine therapy with or without chemotherapy. Premenopausal patients were proved to benefit from chemotherapy, but the benefit was not positively associated with RS. Of note, patients with more than or equal to two positive lymph nodes only took up 34% in this study. Whether the conclusion can be applied to patients with multiple lymph node metastases needs to be further explored [56].

The WSG-ADAPT-HR+/HER2− trial is the initial study to assess the combination of RS and response to neoadjuvant endocrine therapy as an indication of regimens. Ki-67 was tested before and after neoadjuvant endocrine therapy to reflect the response. Patients with RS ranging from 12 to 25 and without endocrine therapy response received adjuvant chemotherapy as the experimental arm, while other patients with RS less than 25 were taken as the control arm. For low-risk patients and patients that respond to neoadjuvant endocrine therapy, endocrine therapy also was not inferior to endocrine therapy plus chemotherapy [57]. In addition, combining RS with response to endocrine therapy is a practicable strategy to guide systemic treatment for HR-positive breast cancer patients with less than three positive lymph nodes [58,59].

Oncotype Dx has been shown to predict the efficacy of adjuvant radiotherapy by analyzing RS and local-regional recurrence (LRR). In the NSABP B-28 study, 10-year LRR increased with the increased risk that was demonstrated by the RS. Further, RS was defined as an independent risk factor of LRR [60]. Another study also revealed that increased RS was associated with increased LRR rates through genetic testing in 316 HR-positive breast cancer patients [61]. Taken together, Oncotype Dx can contribute to the selection of adjuvant radiotherapy by indicating LRR risks in node-positive patients.

### 3.2. MammaPrint

MammaPrint, also known as the 70-gene risk of distant recurrence signature, is an MGA developed for HR-positive HER-2-negative breast cancer patients in stage N0–N1 [62]. Similar to Oncotype Dx, MammaPrint predicts the risk of recurrence and metastasis, as well as indicating treatment management. Using DNA microarray techniques, the MammaPrint tests 70 genes and divides patients into high-risk and low-risk groups. A phase III study, MINDACT, enrolled 6693 breast cancer patients whose genomic risk and clinical risk were assessed through the use of MammaPrint and Adjuvant! Online, respectively. Chemotherapy was prescribed for those who were found to have a high genomic and clinical risk, whereas those with a low risk for both were not given chemotherapy. Patients with controversial genomic risks and clinical risks were randomly assigned to chemotherapy or the control group, and there were no significantly different 5-year distant metastasis-free survival (DMFS) rates between these two groups, indicating that MammaPrint can exempt approximately 46% of clinically high-risk patients from chemotherapy. Patients at both low risks showed the best prognosis, while patients at both high risks benefit from chemotherapy [63]. The results from the 8-year follow-up and subgroup analysis of the MINDACT trial demonstrated that patients who are younger than 50 years old and have distinct genomic and clinical risks may achieve therapeutic effects from chemotherapy [64].

### 3.3. RecurIndex

RecurIndex is employed in N0–N2 stage HR-positive breast cancer patients to direct adjuvant therapy. The 28 genes tested in RecurIndex were established to indicate the risk of distant metastasis in Asian patients, as well as estimate the benefit from adjuvant chemotherapy or radiotherapy. To verify the RecurIndex, a total of 752 operable breast cancer patients were enrolled and divided into high-risk and low-risk groups by RecurIndex. The 10-year relapse-free survival (RFI) for high-risk and low-risk groups was 80.5% vs. 90.0%, and the 10-year distant RFI was 85.0% vs. 94.1%. Subgroup analysis noted a modest chemotherapy benefit in the high-risk group [65]. Another study conducted using 490 HR-positive patients also revealed a significant difference in distant RFI between high-risk and low-risk groups, regardless of lymph node metastasis [66]. The results taken from RecurIndex were proven with prognostic values, which may be conducive to the decision concerning adjuvant chemotherapy.

Further validation research has investigated the predictive role of RecurIndex in adjuvant radiotherapy. A total of 388 patients at clinical stage I–III were followed up, and 10-year local RFI was analyzed. Local RFI was 100% in both the radiotherapy group and the control group for low-risk patients defined by RecurIndex, while radiotherapy improved local RFI at 18.2% for high-risk patients [67]. For stage N1 breast cancer patients, low-risk patients in the RecurIndex test showed no statistical difference in local RFI, distant RFI, recurrence-free survival (RFS), and overall survival (OS) between those who underwent adjuvant radiotherapy and those who did not. High-risk patients who received radiotherapy showed markedly higher distant RFI, local RFI, RFS, and OS [68]. Thus, decision making concerning adjuvant radiotherapy can be guided by RecurIndex, wherein high-risk patients in stage N1 are recommended to undergo radiotherapy to prevent recurrence.

### 3.4. BCI

The BCI was developed for postmenopausal HR-positive node-negative breast cancer patients. To predict prognosis and the response to endocrine therapy, 11 genes, including four reference genes were detected. The results of BCI provide scores for five progression-related genes and the ratio of *HOXB13* to *IL17BR* (H/I), which are genes involved in the estrogen signaling pathway [69,70,71,72]. The Trans-aTTOM trial investigated patients, of which 49% are defined as high risk by BCI (H/I). Compared with patients who received tamoxifen (TAM) for 5 years, patients who received TAM for 10 years showed increased RFI in the high-risk group. In contrast, low-risk patients did not benefit from extended endocrine therapy, demonstrating that patients with elevated BCI can benefit from extended endocrine therapy. The association between BCI and prolonged endocrine therapy stayed present even after eliminating confounding factors, including pathological features [73,74]. The IDEAL study confirmed the benefit of additional letrozole treatment for 5 years compared with an additional 2.5 years. Both clinical high-risk and BCI high-risk patients received better RFI in those who received additional 5-year letrozole. For BCI low-risk patients, no statistical difference was noticed between extending letrozole for 5 years and 2.5 years, regardless of what level of the clinical risk [75]. Extended endocrine therapy was advised for BCI high-risk patients owing to the increased risk of distant recurrence.

### 3.5. EndoPredict

The EndoPredict test combines the expression of 12 genes and clinicopathological features, including tumor size and lymph node metastasis, to rate an EPclin score that indicates prognosis. EPclin score can divide ER-positive HER-2-negative patients into the high-risk group and the low-risk group, hence indicating the risk of recurrence and adjuvant therapy regimens [76]. The ABCSG-6/8 cohorts evaluated EPclin score in postmenopausal ER-positive patients who were administered endocrine therapy only and investigated their distant recurrence-free rate (DRFR). Patients with low EPclin scores showed significantly higher DRFR in both node-positive and node-negative subgroups, validating the prognostic value of EndoPredict [77]. Subsequently, the association of high-risk EPclin scores and worse distant recurrence-free survival (DRFS) in premenopausal patients was revealed via retrospective analysis [78]. The promising predictive role of EndoPredict promoted its exploration in forecasting chemotherapy benefits. A study enrolled 373 ER-positive breast cancer patients with 0–3 metastatic lymph nodes. The 3-year disease-free survival (DFS) was increased by 4.8% (96.3% vs. 91.5%) in high-EPclin score patients who received chemotherapy, confirming the predictive value of EndoPredict in adjuvant chemotherapy benefit [79].

### 3.6. PAM50 Risk of Recurrence (ROR)

The PAM50 assay was designed to classify the intrinsic molecular subtype of breast cancer and differs from other MGAs. It detects 50 oncogenic genes and five reference genes and categorizes breast cancers into luminal A, luminal B, HER-2-enriched, and basal-like subtypes [80]. Potential effective agents were explored for different subtypes in the NCIC.CTG MA.5 trial and NCIC.CTG MA.12 trial, indicating the efficacy of anthracycline for the HER-2-enriched subtype and tamoxifen for luminal subtypes [81,82].

The risk of recurrence (ROR) score was generated, which concluded the results of the PAM50 assay and tumor size. The role of predicting the prognosis that PAM50 subtyping and ROR score possessed was also proved in NCIC-MA.5 and NCIC-MA.12 trials [83]. In a separate trial, patients with low ROR scores and no positive lymph nodes demonstrated optimal outcomes even without receiving adjuvant therapy. This highlights the ROR’s predictive value in prognosis and chemotherapy benefits [84]. The predictive power of ROR, Oncotype Dx, EndoPredict, and BCI was compared in the transATAC study. A total of 785 patients were analyzed, and connections were revealed between the four MGAs. However, Oncotype Dx was found to be stronger in estrogen-related modules, while ROR, BCI, and EndoPredict are more persuasive in proliferative-related genes [85].

Besides results from transATAC, there exist differences between the MGAs for early-stage breast cancer. The applicable populations vary from MGAs. The Oncotype Dx, MammaPrint, BCI, and EndoPredict are applicable for HR-positive HER-2-negative breast cancer patients at stage N0–N1, while RecurIndex is for HR-positive patients at stage N0–N2. PAM50, however, can be used for newly diagnosed breast cancer regardless of molecular subtypes. The genes selected for genetic testing in each MGA are also different. For instance, the status of ER, PR, HER-2, and Ki-67 are not included in the MammaPrint, and whether there exist positive lymph nodes is not stratified. The above information on molecular subtyping was included in the PAM50. Due to the different emphasis of each MGA, the choice of genetic testing should be individualized. Further, RecurIndex was established based on Asian populations, while other MGAs are based on European and American populations. The predictive value across different populations needs further exploration. The gene numbers, applicable populations, and representative trials are summarized, as shown in Table 1.

## 4. Genetic Testing for Metastatic Breast Cancer

Approximately 3–8% of breast cancer patients experience metastasis at the time of initial diagnosis [86]. Metastatic breast cancer is characterized by a poor prognosis, with a 5-year mortality of more than 75% [87,88]. Despite the challenging nature of curing metastatic breast cancer, enhancing current treatment strategies and developing new therapeutic agents could help alleviate symptoms, thereby improving survival rates and quality of life. Studies and comprehension of biomarkers highlight the significance of genetic testing as the basis and prerequisite for precision medicine [89]. The main biomarkers involved include phosphatidylinositol-4,5-bisphosphate 3-kinase, catalytic subunit alpha (*PIK3CA*), estrogen receptor 1 (*ESR1*), *CDK4/6*, *BRCA1/2*, markers for immunotherapy or antibody-drug-conjugates (ADCs), circulating tumor DNA (ctDNA) and circulating tumor cells (CTC), etc. Different genetic testing is recommended for different subtypes of breast cancer, as summarized as follows.

### 4.1. Genetic Testing for Metastatic HR-Positive Breast Cancer

Mutations of *ESR1* and *PIK3CA* are frequently found in HR-positive metastatic breast cancer. *ESR1* mutations are the primary cause of aromatase inhibitor (AI) resistance, which occurs in nearly 30% of HR-positive metastatic breast cancer patients [90,91]. The most commonly identified *ESR1* gene mutations were D538G, Y537S, and Y537N [92]. Trials including SoFEA and EFECT have analyzed the efficacy of fulvestrant compared with AI exemestane in patients who were detected with *ESR1* mutations in ctDNA. Fulvestrant-treated patients demonstrated extended progression-free survival (PFS) in patients who have received AI treatment [93]. The prospective randomized trial PADA-1 further indicated the benefits of fulvestrant. The median PFS of patients who have switched to fulvestrant from letrozole was prolonged for 6.2 months, demonstrating the clinical benefit of fulvestrant for patients with mutated *ESR1* [94].

The PI3K is a key lipid kinase in the PI3K/AKT/mTOR pathway, controlling cell proliferation, metabolism, and other cellular processes. The activation of the PI3K pathway has been illustrated as a bypass that promotes cell proliferation independent of estrogen [95]. Long-term estrogen suppression can result in the activation of PI3K, which has been identified as an antitumor target. In the BELLE-3 trial, combining the PI3K inhibitor buparlisib and fulvestrant achieved longer PFS in HR-positive metastatic breast cancer patients with *PIK3CA* mutations [96]. The results from another clinical trial SOLAR-1 also supported the combination of PI3K inhibitor alpelisib and fulvestrant. The extended PFS in the alpelisib group accelerated the approval of alpelisib for HR-positive metastatic breast cancer patients with *PIK3CA* mutations [97,98]. The most common *PIK3CA* mutations revealed in NGS present in invasive breast cancer are H1047R, E542K, and E545K [99]. Mutations of *PIK3CA* can be detected through the use of the NGS technique, for which tissue from the metastatic site and ctDNA in the plasma can be used to test specimens [100,101,102]. However, the consistency between plasma ctDNA and tumor tissue was poor in the SLOAR-1 trial, and only 177 of 317 patients with mutated *PIK3CA* were detected using plasma ctDNA. Therefore, the retesting of tumor tissue is recommended for patients with no *PIK3CA* mutations found in their ctDNA [103].

Moreover, the combination of CDK4/6 inhibitor and endocrine therapy has emerged as the frontline therapy for HR-positive breast cancer patients. The activation of the CDK4/6 pathway is often observed in various malignancies, driving dysregulated cell cycle and excessive tumor cell proliferation. CDK4/6 inhibitors, including palbociclib, ribociclib, and abemaciclib, have been approved for HR-positive metastatic breast cancer [104]. In the PALOMA-1/TRIO-18 trial, palbociclib plus letrozole led to a prolonged PFS (20.2 months vs. 10.2 months) [105,106]. Comparable favorable outcomes observed in the PALOMA-2 and PALOMA-3 trials reinforced the benefits of combined palbociclib and endocrine therapy [107,108]. A series of MONALEESA trials, including MONALEESA-2, MONALEESA-3, and MONALEESA-7 trials, proved the favored PFS of combining ribociclib with the aromatase inhibitor [109,110,111,112]. Abemaciclib was approved based on its efficacy exhibited in the MONARCH-1, -2, and -3 trials [113,114,115,116]. Resistance to CDK4/6 inhibitors can be encountered in HR-positive metastatic breast cancer. A comprehensive analysis of the genomic profile in metastatic breast cancer reveals a higher prevalence of oncogenic driver gene mutations in patients with HR-positive metastatic breast cancer. The study identifies mutations of nine key oncogenes, namely *TP53*, *ESR1*, *GATA3*, *KMT2C*, *NCOR1*, *AKT1*, *NF1*, and *RB1*, which may contribute to the development of drug resistance [117].

### 4.2. Genetic Testing for Metastatic HER-2-Positive Breast Cancer

Monoclonal antibodies against HER-2, including trastuzumab and pertuzumab, have shown great efficacy for HER-2-positive breast cancer [118]. ADCs combine the precise targeting ability of monoclonal antibodies and the toxicity of chemotherapy drugs, and numerous ADCs have been designed and explored in metastatic breast cancer [119]. As the most commonly used target in breast cancer, ADCs targeting HER-2, including T-DM1 and T-DXd, have been approved. Efficacy was also noted in terms of treating HER-2-low breast cancer with T-DXd [120,121,122].

Mutations of the HER-2 gene may be responsible for drug-resistant HER-2-positive breast cancer. Potential drug resistance mechanism includes incomplete blockade of the HER-2 receptor and activation of compensatory mechanisms within the HER family (HER-3). ADCs targeting HER-3 have been developed and explored [123,124]. The aberrant activation of CDK4/6 and PI3K signaling may also be involved in HER-2-positive metastatic breast cancer [125,126,127]. The BYL-719 study showed that the combination of alpelisib and T-DM1 had better safety and efficacy in patients with trastuzumab-resistant HER-2-positive advanced breast cancer, and the objective response rate (ORR) was higher than that of T-DM1 alone. The ORR was 43%, and the median PFS was 8.1 months [128,129].

### 4.3. Genetic Testing for Metastatic Triple-Negative Breast Cancer (TNBC)

*BRCA1* and *BRCA2* mutations represent the most well-known mutations in breast cancer. They are widely known to drive early-onset breast cancer and frequently occur in metastatic TNBC [130]. Mechanistically, *BRCA1* and *BRCA2* genes encode tumor suppressor proteins, participate in double-stranded DNA homologous recombination repair transcription and cell cycle regulation, and maintain genomic stability. Once the *BRCA1* or *BRCA2* gene is mutated, the tumor suppression effect will be impaired, hence accelerating cancer development and progression [131]. Approximately 20,000 mutations occur within the *BRCA1* and *BRCA2* genes, and the mutation sites are dispersed. Thus, detecting the entire coding region of the *BRCA1/2* gene by high-throughput sequencing is recommended to achieve full coverage of the BRCA1/2 gene in the non-hotspot mutation regions [132,133]. The OlympiAD trial and EMBRACA trial proved the improved efficacy of PARP inhibitors compared with chemotherapy in HER-2-negative metastatic breast cancer patients with *BRCA1/2* mutations. The median PFS in the olaparib group was 3 months longer than the chemotherapy group, and the remission rate was twice as high as that in the chemotherapy group [101,134,135]. Talazoparib also prolonged median PFS for 3 months in patients with germline BRCA mutations [102,136]. Collectively, genetic testing of *BRCA1/2* mutations is recommended for HER-2-negative metastatic breast cancer patients to accurately screen the potential clinical benefit of PARP inhibitors.

In addition, ADCs were developed for TNBC, and emerging targets, including trophoblast cell surface antigen 2 (Trop-2), are being studied. Sacituzumab govitecan (SG), which targets Trop-2, has been approved and shown therapeutic effects for TNBC [137,138]. Therefore, genetic testing concerning the targets of ADCs can be performed in metastatic breast cancer patients to select regimens and predict efficacy.

Diverse genetic mutations were detected in metastatic TNBC. In the LOTUS study, 41% of TNBC were found to carry mutations related to the *PIK3CA* pathway. The addition of the *PIK3CA* inhibitor Ipatasertib significantly benefits these patients [139]. In the FUTURE study, researchers conducted comprehensive genetic testing of advanced drug-resistant TNBC, and the results showed that the most commonly mutated genes in refractory TNBC included *TP53* (72%), *PIK3CA* (18%), *PTEN* (10%), *KMT2D* (9%), and *TSC* (29%) [140]. Inhibitors of these molecules may also achieve therapeutic effects in metastatic TNBC patients.

### 4.4. Genetic Testing for Immunotherapy

By eliciting the antitumor immune response of the host, immunotherapy has been one of the most prospective treatments recently. The PD-L1 expression, TMB, and mismatch repair deficiency (dMMR) are deemed biomarker candidates for immunotherapy [141,142].

The approach to detect PD-L1 expression is not uniform, and combined positive score (CPS) is conducted the most widely [143]. CPS refers to the percentage of cells that can be positively stained by PD-L1 antibodies in immunohistochemistry tests in all alive tumor cells [144]. The results of the KEYNOTE-355 trial supported that PD-L1-positive patients can better benefit from pembrolizumab [145]. High levels of TMB indicate more mutations in cancer genomes and more neoantigens, hence strengthening the antitumor response [146]. The connection between TMB and sensitivity to immunotherapy was proved in the TAPUR study, in which the objective response rate (ORR) to pembrolizumab was 37% in patients with high TMB [147]. dMMR also results in increased mutations that cause microsatellite instability, making patients potentially respond to anti-PD-L1 therapy [148,149]. Promising response rates and extended response durations were observed in the KEYNOTE-158 trial when dMMR patients with solid tumors were treated with anti-PD-L1 agents [150]. The detection approaches of dMMR include immunohistochemistry tests, PCR assays, or NGS. In addition, the detection of dMMR and TMB can be coupled in NGS tests, making NGS the decisive tool [151]. Although TNBC was regarded to be more sensitive to immunotherapy, all subtypes of metastatic breast cancer would be recommended to undergo genetic testing for immunotherapy to access a potential regimen [152,153,154].

### 4.5. Potential Genetic Testing for Monitoring Efficacy

Biomarkers, including ctDNA and CTCs, can be detected via genetic testing to monitor therapeutic response in metastatic breast cancer patients. ctDNA, a distinctive tumor biomarker, comprises mutated gene fragments secreted by cancer cells. ctDNA can be detected via PCR or sequencing the plasma of patients [155]. The identification of ctDNA holds as a means of detecting tumor-specific mutations in metastatic breast cancer. It is believed that ctDNA testing, as an important liquid biopsy technology, can be used as an alternative method for tissue biopsy. Genetic testing of ctDNA effectively evaluates the TMB and molecular characteristics of advanced breast cancer and has certain clinical value in selecting effective treatment methods and dynamic monitoring therapeutic response [156]. The accuracy of ctDNA testing was verified through the plasmaMATCH trial that compared the capacity of ctDNA testing and testing in biopsy tissues. Tissue sequencing was set as the golden standard, and sensitivities of 93% and 98% were observed in ctDNA testing and biopsy testing, respectively. In addition, in patients with HER-2 and *AKT1* mutations, targeting mutations detected in ctDNA could achieve considerable therapeutic effects [157]. The ctDNA levels are closely associated with tumor burden, and increased ctDNA levels may indicate disease progression [158,159]. Given that ctDNA testing is noninvasive, rapid, cost-effective, and accurate, it possesses the potential for clinical application that is worth exploring [160].

The CirCe01 trial was designed to assess whether CTC monitoring can improve the survival of metastatic breast cancer patients receiving chemotherapy. Before starting the first cycle of chemotherapy, patients with ≥5 CTCs/7.5 mL were randomly assigned to the CTC group and the standard group. The CTC group received CTC monitoring at each subsequent cycle of chemotherapy, while patients in the standard group were treated according to imaging assessment every three cycles. The results showed no significant difference between the two groups, indicating the insufficiency of recommending CTCs to monitor treatment response in patients with metastatic breast cancer [161,162]. However, the value of CTC testing for prognostic and prediction purposes in breast cancer was validated [163]. Blood CTC levels of >1 CTC/7.5 mL were linked with a more than 12 times higher risk of recurrence in HR-positive breast cancer [164]. The predictive role of CTCs was explored in patients who received radiotherapy. In patients with detectable CTCs, radiotherapy effectively prolonged their DFS and OS [165]. Together, more studies on the application of ctDNA testing and CTC testing for breast cancer are warranted.

Genetic testing for metastatic breast cancer is mainly based on therapeutic targets. Different biomarkers indicate different drugs or treatment strategies. In addition, less traumatic, easily accessed testing, including ctDNA and CTC, is being investigated to monitor efficacy. Compared with genetic testing for early-stage breast cancer, genetic testing for metastatic breast cancer tends to evaluate the efficacy of specific drugs instead of the benefits of the treatment strategy. In addition, the results of MGAs for early-stage breast cancer are presented as a score rather than positive or negative. Genetic testing for different subtypes of metastatic breast cancer is shown in Table 2.

## 5. Advantages and Limitations of Genetic Testing

Genetic testing provides a promising future for precision medicine. Currently, breast cancer is divided into HR-positive, HER-2-positive, and TNBC subtypes according to the expression status of ER, PR, and HER-2. The management of treatment is mainly based on the molecular subtypes and clinical stages. With a deeper comprehension of molecular pathology, genetic testing has been designed and explored. Genetic testing enables more precise risk stratification of breast cancer patients, indicating the benefit of chemotherapy and the risk of recurrence. In addition, genetic testing helps to recognize therapeutic targets of metastatic breast cancer patients and reveals hereditary risk. Comprehensive, precise, and personalized treatment can be conducted due to evidence from genetic testing.

However, limitations remain in the implementation and advancement of genetic testing for breast cancer. Firstly, ethnic disparities may impact MGA testing outcomes. For Chinese patients, only the RecurIndex has been studied in the Chinese population, while the rest of the MGAs are based on the detection and validation in European and American populations [166,167,168]. The impact of ethnic differences on test results remains unclear since the genetic backgrounds used to establish the MGAs are different [169]. Secondly, similar problems are faced when conducting NGS in metastatic breast cancer patients. Bioinformatics analysis of NGS results utilizes the gene–population database, gene–disease, and gene–drug association database [170,171,172], which primarily originated in Western populations. Genetic diversity across various ethnic groups may result in dissimilar gene mutations, rendering the databases unsuitable for other populations. The reported germline BRCA1/2 mutations in Chinese patients with hereditary breast and ovarian cancer are highly ethnically specific [173,174]. Some studies have shown that the differences in the mutation spectrum of breast cancer between different countries are also clustered within HR-negative and HER-2-negative subtypes [175]. Unique disease characteristics were identified in premenopausal breast cancer patients in Asia, indicating potential benefits from regimens that deviate slightly from international guidelines [176]. Thirdly, decision making in clinical practice requires a combination of clinicopathological features and genetic testing results. Although MGAs show reliability in validation trials, decisions regarding adjuvant therapy for breast cancer still depend on accurate clinicopathological features. In some cases, the indications from the MGAs can be controversial in terms of such features [177]. Fourthly, genetic testing products should be qualified, and their reliability should be confirmed. It is advisable to use original research products or qualified testing organizations for multi-gene testing. However, the relatively high economic cost of genetic testing makes the promotion of MGAs limited [178,179,180].

## 6. Conclusions and Future Directions

In this review, we summarized genetic testing for breast cancer, encompassing both early-stage and metastatic cases. Based on evidence from genetic testing and clinicopathologic characteristics, physicians and pathologists could have a more sophisticated comprehension of managing breast cancer, hence stepping precision medicine forward. For patients in the early stage, MGAs including Oncotype Dx, MammaPrint, RecurIndex, BCI, EndoPredict, and PAM50 aid in assessing and predicting prognosis as well as identifying potential benefits from chemotherapy or extended endocrine therapy. Mutations of important oncogenic drivers, including *ESR1*, *PIK3CA*, *CDK4/6*, *BRCA1/2*, *TP53*, *PTEN*, and expressions of genes indicating immunotherapy or ADC targets contribute to selecting effective agents for metastatic breast cancer patients. However, genetic testing poses challenges in the rapidly evolving field of precision medicine. Clinical trials should be conducted in diverse races to further convince evidence from genetic testing, as well as explore new profiles of genes in different subtypes of breast cancer. Advancements in novel tools and techniques are necessary for genetic tests to be accessible and cost-effective. Altogether, genetic testing holds promise for advancing the precise management of breast cancer.

## Figures and Tables

**Figure 1 ijms-24-16607-f001:**
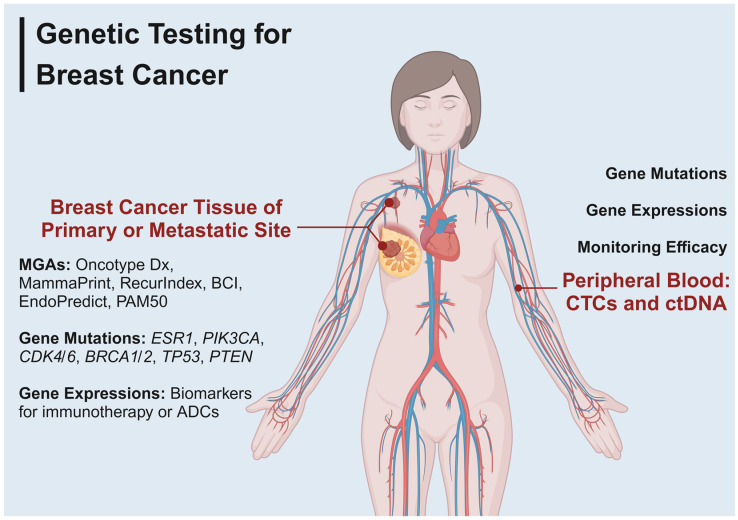
Genetic testing for breast cancer. ADCs, antibody-drug conjugates; CTCs, circulating tumor cells; ctDNA, circulating tumor DNA.

**Table 1 ijms-24-16607-t001:** Genetic testing for early-stage breast cancer.

MGAs	Gene Number	Detection Method	Applicable Population	Representative Trials	Applications
Oncotype Dx	21	RT-PCR	HR-positive HER-2-negative	NSABP B-14	Predict prognosis,
			breast cancer, Stage N0–N1	NSABP B-20	Direct adjuvant chemotherapy or radiotherapy
				TAILORx	
				SWOG S8814	
				RxPONDER	
				WSG-ADAPT-HR+/HER2−	
				NSABP B-28	
MammaPrint	70	DNA-Microarray	HR-positive HER-2-negative breast cancer, Stage N0–N1	MINDACT	Predict the risk of recurrence and metastasis
RecurIndex	28	RT-PCR	HR-positive HER-2-negative breast cancer, Stage N0–N2	Validation Researches	Direct adjuvant chemotherapy or radiotherapy
BCI	11	RT-PCR	HR-positive HER-2-negative	Trans-aTTOM	Predict prognosis,
			breast cancer, Stage N0–N1	IDEAL	Predict response to endocrine therapy
EndoPredict	12	RT-PCR	HR-positive HER-2-negative	ABCSG-6	Predict prognosis
			breast cancer, Stage N0–N1	ABCSG-8	
PAM50	55	RT-PCR nCounter	Newly diagnosed breast cancer	NCIC.CTG MA.5	Classify the intrinsic molecular subtype,
				NCIC.CTG MA.12	Predict the risk of recurrence
				transATAC	

**Table 2 ijms-24-16607-t002:** Genetic testing for metastatic breast cancer.

Subtype	Genetic Testing	Gene	Potential Drugs	Representative Trials
HR-positive	Gene Mutations	*ESR1*	Fulvestrant	SPFEA
				EFECT
				PADA-1
		*PIK3CA*	PI3K Inhibitors	BELLE-3
				SOLAR-1
	Gene Expressions	*CDK4/6*	CDK4/6 Inhibitors	PALOMA
				MONALEESA
				MONARCH
HER-2-positive	Gene Expressions	HER-2	T-DM1	EMILIA
			T-DXd	DESTINY-Breast 03
		HER-3	HER3-DXd	NCT02980341
				NCT04610528
	Gene Mutations	PI3K	PI3K Inhibitors	BYL-719
TNBC	Gene Mutations	*BRCA1/2*	PARP Inhibitors	OlympiAD
				EMBRACA
		*PI3CA*	PI3K Inhibitors	LOTUS
		*TP53*	Inhibitors	FUTURE
		*PTEN*	Inhibitors	FUTURE
	Gene Expressions	Trop-2	Sacituzumab Govitecan	ASCENT
All Subtypes	Biomarkers for Immunotherapy	PD-L1 Expression	ICIs	KEYNOTE-355
		TMB		TAPUR
		dMMR		KEYNOTE-158
	Monitoring Efficacy	ctDNA	-	plasmaMATCH
		CTCs	-	CirCe01

Abbreviations: HR, hormone receptor; HER-2, human epidermal growth factor receptor-2; TNBC, triple-negative breast cancer; TMB, tumor mutational burden; dMMR, mismatch repair deficiency; ICIs, immune checkpoint inhibitors; ctDNA, circulating tumor DNA; CTCs, circulating tumor cells.

## Data Availability

No new data were created or analyzed in this study. Data sharing is not applicable to this article.

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
