# Peer review of "Genetic Testing Enhances the Precision Diagnosis and Treatment of Breast Cancer"

_ijms, 2023, doi:10.3390/ijms242316607_

Round 1

Reviewer 1 Report

Comments and Suggestions for Authors

The manuscript "Genetic Testing Enhances the Precision Diagnosis and Treatment of Breast Cancer" is relevant to the International Journal of Molecular Sciences journal. The article presents the development of precision diagnosis and treatment of breast cancer in an interesting way. Many literature references, especially from recent years, testify to a deep analysis of the described topic. In my opinion, there is nothing to complain about. However, I have some questions and remarks.

Line 108 – Are CDH1, ATM, CHEK2 and PALB2 also biomarkers? This is the first time it has been mentioned.

Line 126 – It can be written (summarized) that the following describes MGAs, and only at the end write about the summary in Table 1. Now, there is a reference to Table 1, which represents the MGAs discussed in detail later in the chapter.

Line 206 and 207 – There should be an abbreviation vs. not v.s. Shortcut v.s. (with a dot after each letter) is an abbreviation of vide supra 'see above' (a note in the text referring to the content previously included in the text). Please read the entire text and correct it.

Line 244 – It should be HR, not ER. Please explain if this is correct because it is also repeated on line 247.

Line 248 – There is ERclin, and a few lines higher, there is EPclin. Which form is correct?

Line 254 – What is DFS? This has not been explained before.

Line 275 – Please refer to Table 1 here, not at the beginning of the chapter when the readers don't know what it is about.

Table 1 – Why are MGAs: BCI and EndoPredict not separated? Were they used in the same repressive trials?

Line 288 – As in Chapter 3, I would refer to Table 2 at the end of this chapter. I would even use this sentence to summarise this chapter and reference Table 2, where all the things described above were presented.

Line 300 – Time unit. Please forgive me, but this is the first time I have come across such symbols as 6.2, 20.2, 10.2 or 8.1 months. Wouldn't it be better to convert it into the number of days?

Line 327 – MONALEESA trials are listed. However, trials 4-6 are missing in the series calculation. Why? Were they about something else?

Line 441 – There should be a summary and reference to Table 2.

Comments on the Quality of English Language

Please use English that can be understood by technical and non-technical readers. Check language once more carefully. Also, consider a native speaker or professional help to ensure submission of the highest possible language level that authors really care about.

Author Response

Dear reviewer,

We feel great thanks for your professional comments concerning our manuscript “Genetic Testing Enhances the Precision Diagnosis and Treatment of Breast Cancer”. The comments are all valuable and very helpful for further improving our paper. We read the comments carefully and have made corresponding corrections which we hope to meet with your approval.

Yours sincerely,

Tinglin Yang, Wenhui Li, Tao Huang, and Jun Zhou

The main corrections in the paper and the response to your comments are as follows. All modifications in the manuscript have been marked up by using the “track changes” function in MS Word.

Comment #1: The manuscript "Genetic Testing Enhances the Precision Diagnosis and Treatment of Breast Cancer" is relevant to the International Journal of Molecular Sciences journal. The article presents the development of precision diagnosis and treatment of breast cancer in an interesting way. Many literature references, especially from recent years, testify to a deep analysis of the described topic. In my opinion, there is nothing to complain about. However, I have some questions and remarks.

Line 108 – Are CDH1, ATM, CHEK2 and PALB2 also biomarkers? This is the first time it has been mentioned.

Response #1: CDH1, ATM, CHEK2, and PALB2 (Line 137) are all genes whose mutations can be oncogenic. Similar to BRCA1/2, TP53, PETN, and STK11 mentioned above in the same paragraph, mutations in these genes increase hereditary risks of breast cancer. Genetic testing of these genes can be conducted for assessing hereditary risks of breast cancer.

Comment #2: Line 126 – It can be written (summarized) that the following describes MGAs, and only at the end write about the summary in Table 1. Now, there is a reference to Table 1, which represents the MGAs discussed in detail later in the chapter.

Response #2: We have revised these sentences according to your suggestion (Line 207). The reference and context of Table 1 were moved to the end of section 3 (Line 448).

Comment #3: Line 206 and 207 – There should be an abbreviation vs. not v.s. Shortcut v.s. (with a dot after each letter) is an abbreviation of vide supra 'see above' (a note in the text referring to the content previously included in the text). Please read the entire text and correct it.

Response #3: Thank you for pointing out our mistakes. The abbreviation vs. was corrected in our whole manuscript (Line 293, Line 294, Line 361, Line 508).

Comment #4: Line 244 – It should be HR, not ER. Please explain if this is correct because it is also repeated on line 247.

Response #4: In breast cancer, hormone receptors (HR) include estrogen receptor (ER) and progesterone receptor (PR), and ER is the most important classification criteria and therapeutic target. Since these trials included patients who were all ER-positive, we used the term ER rather than HR.

Comment #5: Line 248 – There is ERclin, and a few lines higher, there is EPclin. Which form is correct?

Response #5: We apologize for the typos. It is EPclin, and was corrected (Line 354).

Comment #6: Line 254 – What is DFS? This has not been explained before.

Response #6: Sorry for our carelessness. The full name “disease-free survival” for DFS was added (Line 360).

Comment #7: Line 275 – Please refer to Table 1 here, not at the beginning of the chapter when the readers don't know what it is about.

Response #7: The reference of Table was moved close to it (Line 447) according to your suggestion.

Comment #8: Table 1 – Why are MGAs: BCI and EndoPredict not separated? Were they used in the same repressive trials?

Response #8: We apologize again for our carelessness. BCI and EndoPredict was separated (Line 448).

Comment #9: Line 288 – As in Chapter 3, I would refer to Table 2 at the end of this chapter. I would even use this sentence to summarise this chapter and reference Table 2, where all the things described above were presented.

Response #9: Thank you for your suggestion. We moved the reference to Table 2 to the end of the section 4 (Line 692).

Comment #10: Line 300 – Time unit. Please forgive me, but this is the first time I have come across such symbols as 6.2, 20.2, 10.2 or 8.1 months. Wouldn't it be better to convert it into the number of days?

Response #10: Thank you for pointing out that the time units in our manuscript are not uniform. However, since the units given in the original clinical trials are months, we think it may bring unnecessary errors to convert them into days, so we still decided to use the original time units in this manuscript. Thank you again for your kindness.

Comment #11: Line 327 – MONALEESA trials are listed. However, trials 4-6 are missing in the series calculation. Why? Were they about something else?

Response #11: It is indeed an interesting concern. The MONALEESA trials-2,3,7 are the most famous trials that revealed the efficacy of CDK4/6 inhibitors. We searched MONALEESA trials in clinical information disclosure databases of America, China, the European Union, the World Health Organization, Japan, and India. (www.clinicaltrials.gov; http://www.chinadrugtrials.org.cn/index.html; https://www.clinicaltrialsregister.eu/ctr-search/search; https://trialsearch.who.int; https://www.clinicaltrials.jp/cti-user/trial/Search.jsp; https://ctri.nic.in/Clinicaltrials ). However, no information was found about MONALEESA-4-6 trials. We will keep constant attention on MONALEESA trials in our future work.

Comment #12: Line 441 – There should be a summary and reference to Table 2.

Response #12: We summarized and compared genetic testing mentioned in our manuscript at the end of section 4. Besides, the reference to Table 2 were moved to here (Line 692).

Comments on the Quality of English Language: Please use English that can be understood by technical and non-technical readers. Check language once more carefully. Also, consider a native speaker or professional help to ensure submission of the highest possible language level that authors really care about.

Response: We really appreciate your valuable comments and patience. We tried our best to polish the manuscript and made revisions that would not influence the content and framework of our paper. We have also invited a professional native speaker to polish our manuscript, which we hope to meet with your approval.

Reviewer 2 Report

Comments and Suggestions for Authors

The manuscript treats an exciting subject and considers the latest advances in the field.

The authors are kindly requested to consider the following recommendations: 

- make a short introductory presentation of each considered genetic testing;

- present the advantages and limitations of the tests;

- move Table 1 close to where it is referenced in the text;

- include a comparison of the early-stage genetic testings, as well as the metastatic ones;

- if possible, compare the early and metastatic genetic testings in terms of their performances;

- make a short comparison to other used methods.

Comments on the Quality of English Language

Minor English editing.

Author Response

Dear reviewer,

We feel great thanks for your professional comments concerning our manuscript “Genetic Testing Enhances the Precision Diagnosis and Treatment of Breast Cancer”. Those comments are all valuable and very helpful for further improving our paper. We read the comments carefully and have made corresponding corrections which we hope to meet with your approval.

Yours sincerely,

Tinglin Yang, Wenhui Li, Tao Huang, and Jun Zhou

The main corrections in the paper and the response to your comments are as follows. All modifications in the manuscript have been marked up by using the “track changes” function in MS Word.

Comment #1: The manuscript treats an exciting subject and considers the latest advances in the field. The authors are kindly requested to consider the following recommendations: make a short introductory presentation of each considered genetic testing;

Response #1: Thank you for your suggestion. We introduced each genetic tests at the beginning of the paragraph. According to your comment, we tried to revise the introduction to make it more impressive.

Comment #2: present the advantages and limitations of the tests;

Response #2: The advantages and limitations of tests are concluded in section 5 (Line 698-741). Besides, to make the manuscript more logical and accurate, we revised the title of section 2 to “The applications of genetic testing” (Line 80).

Comment #3: move Table 1 close to where it is referenced in the text;

Response #3: We moved the reference of Table 1 and Table 2 to the end of each section (Line 416, Line 655), and tables are placed next to the reference. (Line 448, Line 693)

Comment #4: include a comparison of the early-stage genetic testings, as well as the metastatic ones;

Response #4: To compare genetic testing is indeed important to further introduce them. The comparison of MGAs is added (Line 434-Line 447), as well as a new column of their applications in Table 1 (Line 448). For the metastatic ones, please see Line 663-692.

Comment #5: if possible, compare the early and metastatic genetic testings in terms of their performances;

Response #5: The comparison of early and metastatic genetic testing is introduced along with the comparison of metastatic ones (Line 663-692).

Comment #6: make a short comparison to other used methods.

Response #6: In clinical practice, breast cancer is divided into HR-positive, HER-2-positive, and TNBC subtypes according to the expression status of ER, PR, and HER-2. The comparison of genetic testing to such conventional classification is where its advantage lies (Line 698-706). In addition, due to the limitations of our knowledge, if you know of other genetic testing methods that can help enrich our manuscript, please also let us know, which will make our manuscript more adequate.

Comments on the Quality of English Language: Minor English editing.

Response: We appreciate your valuable comments and patience. We tried our best to polish the manuscript and made revisions that would not influence the content and framework of our paper. We have also invited a professional native speaker to polish our manuscript, which we hope to meet with your approval.

Round 2

Reviewer 2 Report

Comments and Suggestions for Authors

The authors have addressed all of the recommendations.